# Ghost-ResNeXt: An Effective Deep Learning Based on Mature and Immature WBC Classification

**Sai Sambasiva Rao Bairaboina**  **and Srinivasa Rao Battula** *

School of Computer Science and Engineering, VIT-AP University, Amaravati 522237, India; sambasiva.20phd7016@vitap.ac.in
* Correspondence: srinivas.battula@vitap.ac.in

**Abstract:** White blood cells (WBCs) must be evaluated to determine how well the human immune system performs. Abnormal WBC counts may indicate malignancy, tuberculosis, severe anemia, cancer, and other serious diseases. To get an early diagnosis and to check if WBCs are abnormal or normal, one needs to examine the numbers and determine the shape of the WBCs. To address this problem, computer-aided procedures have been developed because hematologists perform this laborious, expensive, and time-consuming process manually. Resultantly, a powerful deep learning model was developed in the present study to categorize WBCs, including immature WBCs, from the images of peripheral blood smears. A network based on W-Net, a CNN-based method for WBC classification, was developed to execute the segmentation of leukocytes. Thereafter, significant feature maps were retrieved using a deep learning framework built on GhostNet. Then, they were categorized using a ResNeXt with a Wildebeest Herd Optimization (WHO)-based method. In addition, Deep Convolutional Generative Adversarial Network (DCGAN)-based data augmentation was implemented to handle the imbalanced data issue. To validate the model performance, the proposed technique was compared with the existing techniques and achieved 99.16%, 99.24%, and 98.61% accuracy levels for Leukocyte Images for Segmentation and Classification (LISC), Blood Cell Count and Detection (BCCD), and the single-cell morphological dataset, respectively. Thus, we can conclude that the proposed approach is valuable and adaptable for blood cell microscopic analysis in clinical settings.

**Keywords:** image classification; white blood cell; deep learning; W-Net; GhostNet



## 1. Introduction

Blood is an essential component of life in the human body. Blood cells and plasma are the components of human blood. Most of the blood is made up of a yellow liquid called plasma, which makes up about 55% of the blood volume. Blood also contains blood cells, hormones, carbon dioxide, proteins, carbohydrates, and micronutrients [1–3]. The three major biological elements of blood are white blood cells (WBCs), platelets (thrombocytes), and red blood cells (RBCs), which can be individually identified by their color, structure, and shape. Neutrophils, eosinophils, basophils, monocytes, and lymphocytes are the five types of WBCs.

WBC analysis is critical and considerably aids in the tracking and early detection of our immune status. Additionally, it can inform a diagnosis for conditions including HIV, acute myeloid leukemia (AML), and adult acute lymphoblastic leukemia (ALL). An essential characteristic of AML is rapid growth in the number of immature blood cells. These cells replace healthy blood cells, proliferate, and prevent the production of new healthy cells from the bone marrow [4–7]. Therefore, the initial stage in the diagnosis of AML is the detection of immature WBCs. In this detection, WBCs are classified as healthy or immature cells and divided into subgroups. The classification of immature WBCs is highly complicated because of their intricacy and resemblance to each other.

The typical method for diagnosing leukemia involves microscopy of peripheral blood smears, though alternative techniques are also employed. Traditional blood smear examination is a lengthier and more effort-intensive process [8–11]. Further, such examination is vulnerable to several errors, including fatigue, operator experience, and significant inter- and intra-observer variation of standards. Manual inspection has a 30–40% error rate, based on the hematologist's experience level. Hence, an efficient technique is necessary for the fast diagnosis of the illness [12–14]. Challenges in the existing manual technique of identification can be overcome by an automated system, especially in the poorest countries, allowing for the standardized and effective detection of mature and immature WBCs.

The automatic categorization of WBCs using computer vision algorithms has drawn significant scientific attention in digital image processing technology [15–17]. However, the categorization and identification of WBCs based on machine learning are difficult because of the morphological similarity among several subtypes and their structural abnormalities. In modern medical imaging, deep learning-based techniques are the most effective for identification and categorization tasks [18–20]. Moreover, they perform very well on large datasets.

This paper classified normal WBCs into five categories and immature ones into seven categories. An efficient segmentation and feature extraction was carried out for this purpose using W-net and GhostNet. Thereafter, ResNeXt-based multi-class classification on WBC was carried out with the aid of Wildebeest Herd Optimization (WHO). Deep Convolutional Generative Adversarial Network (DCGAN)-based image augmentation was also employed to address the issues of imbalanced data and a lack of appropriate sample sizes.

The key contributions of this study are summarized below:

(1) To improve the classifier's overall performance, we suggest a powerful W-net-based segmentation that can precisely find the WBC area.
(2) We developed an efficient GhostNet-based deep learning technique to gather all high-level and low-level features of WBCs.
(3) We employed ResNeXt with WHO to effectively classify atypical WBCs, including immature WBCs.
(4) A data augmentation strategy based on DCGAN was applied to increase the number of images in the dataset and to train the deep learning model successfully.
(5) Various public datasets were used in extensive trials to show how effectively the proposed approach outperforms other well-known methods.

The rest of the paper is structured as follows. Section 2 discusses earlier research on WBC segmentation and classification. Section 3 explains the proposed methodology. Section 4 contains the experimental findings and their implementations, and Section 5 includes the conclusion.

## 2. Literature Review

Ahmad et al. [21] suggested a hybrid strategy for classifying WBCs and first preprocessed the input dataset before inputting it via feature extraction and feature fusion, utilizing a transfer learning step. This was done by using DarkNet53 and DenseNet201—two deep learning methods. An entropy-controlled Marine Predators Algorithm was then used to choose the best attributes. In terms of several performance indicators, this strategy was compared with other existing methods.

A two-stage deep learning model was developed by Elhassan et al. [22] to categorize abnormal WBCs. To create different synthetic WBC pictures, an initial augmentation technique based on geometric modification and the Deep Convolutional Auto-Encoder (DCAE) generating model was presented. Further, the WBCs were divided by utilizing the framework of CMYK-Moment Localization-Feature Fusion Extraction. Then, binary and multi-class classifications were carried out using a hybrid DCAE/Convolutional Neural Network (CNN). Using DCAE, the image was first transformed into a refined version. Then it was supplied into the CNN for further extraction of features. The model's outputs were compared against existing techniques to classify abnormal WBCs.

Cheuque et al. [23] classified four blood cell categories, including eosinophils, neutrophils, monocytes, mononuclear, and lymphocytes, using a multi-level hybrid system. A Faster R-CNN network was used at the first level to identify the region of interest in WBCs and to distinguish polymorphonuclear from mononuclear cells. On being separated, the second-level subclasses were recognized using two adjacent CNNs in the MobileNet framework. The suggested model presented the results of Monte Carlo cross-validation according to F1-score, precision, recall, and accuracy.

Using a random forest approach, Dasariraju et al. [24] created a model that could precisely identify and divide immature leukocytes into four categories: myeloblasts, promyelocytes, monoblasts, and erythroblasts. To acquire the masking of the cytoplasm and nuclei, a segmentation method based on multi-Otsu thresholding was used. The feature extraction approach was used to obtain 16 cytomorphological characteristics. Finally, using the retrieved features as a basis, the random forest algorithm performed the classification where accuracy, precision, recall, and specificity measurements were applied to evaluate performance.

For WBC segmentation, Akram et al. [25] developed a multi-scale information fusion network (MIF-Net) with external and internal procedures for fusing spatial information; MIF-Net is a deep network. To fuse external information, the MIF-Net separated and reproduced the boundary information on multiple scales. For internal fusion, the splitter also transmitted spatial data. It ensured network-wide feature empowerment. After going through a few stages of processing, internal information fusion was finally combined with the fusion of external data. Outcome masks were ultimately created using fused characteristics. On four publicly accessible datasets, the performance assessment was conducted.

Haider et al. [26] presented two deep models for the combined segmentation of nuclei and cytoplasm in WBC pictures: the leukocyte deep aggregation segmentation network (LDAS-Net) and the leukocyte deep segmentation network (LDSNet). For feature extraction, only three down-sampling stages were included in the LDS-Net. To minimize the loss of spatial information and transmit low-level information, the improved version of the LDS-Net-LDAS-Net was presented. To combine low-level data with downscaled spatial features in the LDAS-Net, a dense feature concatenation block was used.

Finally, the overall summary of the literature review is presented in Table 1.

**Table 1.** Summary of literature review.

| Reference | Technique | Dataset | Performance Metrics | Drawback |
|---|---|---|---|---|
| Ahmad et al. [21] | DarkNet53 and DenseNet201 | Real-world large-scale dataset | Accuracy, ANOVA test | This framework is computationally expensive due to the two feature extractors and feature fusion process |
| Elhassan et al. [22] | deep convolutional autoencoder (DCAE) | Single-cell morphological dataset | Precision, sensitivity, and F1-score | For feature extraction, the entire input image was used. As a result, undesirable traits were also extracted from the image |
| Cheuque et al. [23] | MobileNet | Blood Cell Detection, Complete Blood Count dataset, White Blood Cells dataset, Kaggle Blood Cell Images dataset, LISC dataset | Accuracy, recall, precision, and F1-score | Overfitting occurs due to the imbalanced dataset problem |
| Dasariraju et al. [24] | random forest | Single-cell morphological dataset | accuracy, recall, precision, and specificity | Precision value of Promyelocyte class is very low. This means the technique attains a higher number of false positives due to imbalanced classes |

**Table 1.** *Cont.*

| Reference | Technique | Dataset | Performance Metrics | Drawback |
|---|---|---|---|---|
| Akram et al. [25] | MIF-Net | Four private datasets | Precision, misclassification error, dice coefficient, mean intersection over union, false positive rate and false-negative rate | They do not perform any pre-processing techniques to improve the quality and remove noise from the image. This reduces the learning capacity of the network |
| Haider et al. [26] | LDAS-Net | Four private datasets | Precision, misclassification error, dice coefficient, mean intersection over union, false positive rate and false-negative rate | Computationally expensive |

## 3. Proposed Methodology

The created framework entailed five steps: picture pre-processing, image augmentation, WBC segmentation, feature extraction, and classification. Initially, the input images were obtained from the dataset, and then the pre-processing and data augmentation using DCGAN were performed. Afterward, W-Net-based deep learning technique was used to segment the WBC nuclei. Then, the significant features from the segmented images were extracted by the GhostNet-based technique. Finally, the extracted features were sent to the optimized ResNeXt classifier to perform the multi-class classification on mature and immature WBCs. To improve the performance of the classifier, WHO-based hyperparameter optimization was applied. Figure 1 depicts the proposed framework's overall structure.

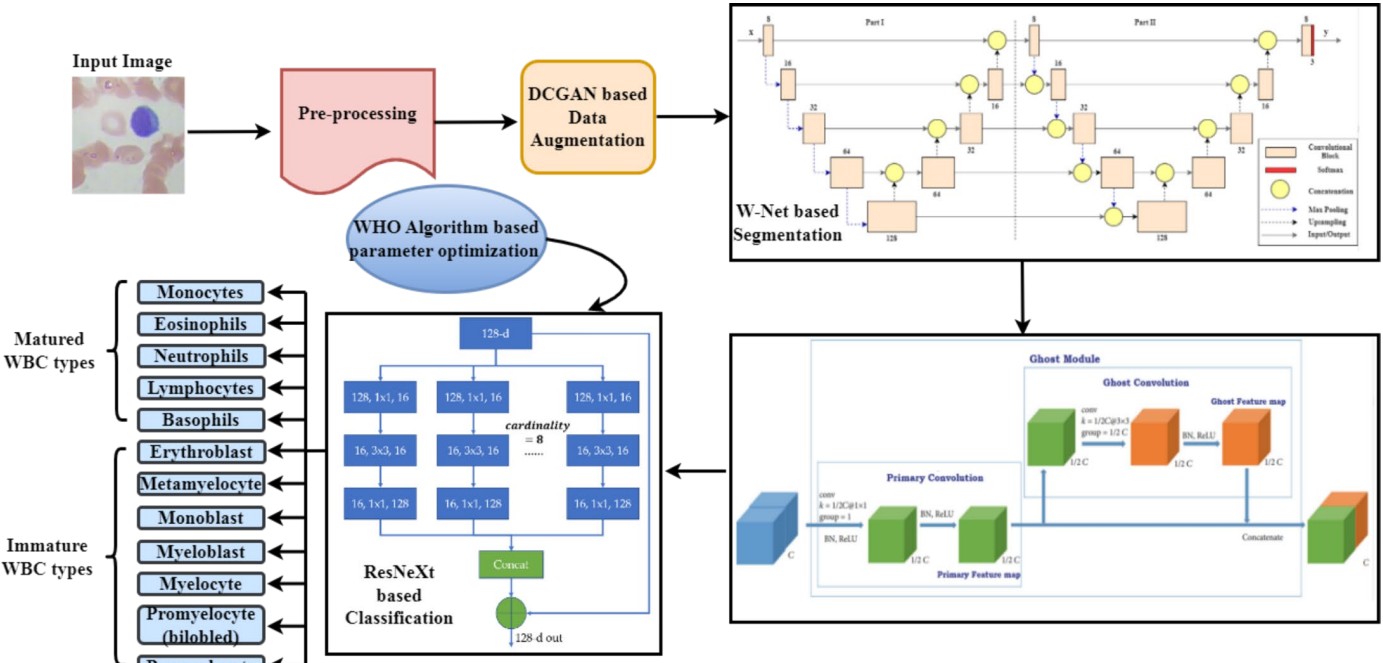

**Figure 1.** System architecture of the proposed framework.

### 3.1. Pre-Processing

As a precondition, the images should be ready for a successful outcome before their analysis. Therefore, image pre-processing is crucial in examining the data for experiments. Each photograph was resized into a 256 × 256 resolution image in this work. After that, the complete dataset was subjected to image contrast enhancement based on Contrast Limited Adaptive Histogram Equalization (CLAHE) to boost contrast and create cell bodies more visible.



### 3.2. DCGAN-Based Data Augmentation

This section presents the DCGAN-based model for oversampling a few rare classes, such as promyelocyte (bilobed), metamyelocyte, etc. This network produced more than 10,000 image data to oversample these rare classes. For both the discriminator and generator models in this network, convolutional neural layers were utilized. These layers were used to produce a more stable architecture and obtain improved outcomes. Compared to traditional GANs, it produces more images with better quality. In addition, it performs more consistently while learning. This network was organized by the following objective function:

$$V(D, G) = E_x \sim p_{data(x)}[\log D(x)] E_z \sim p_{z(z)}[\log(1 - D(G(z)))] \tag{1}$$

Here, the real sample is denoted by x. D(G(z)) represents the probability that the discriminator network D would identify. G(z) is a real sample. D(x) denotes the probability that the discriminator networks would correctly recognize x as a real sample. A sample produced by the generator network G from noise z is denoted by G(z).

In the generator and discriminator network, stride convolution substituted the pooling operation in DCGAN. Additionally, global pooling was used in place of the fully connected layer to increase model stability. Then, using the following (Equations (2) and (3)), the discriminator loss L(D) and generator loss L(G) were determined.

$$L(G) = \frac{1}{N} \sum_{i=1}^{N} -\log(D(G(z_i))) \tag{2}$$

$$L(D) = \frac{1}{N} \sum_{i=1}^{N} -\log(D(x_i)) - \log(1 - D(G(z_i))) \tag{3}$$

Adam optimizer updated the producing network and discriminated network parameters during the training phase, based on the loss functions mentioned earlier. After the first layer of convolution, the combination mode of convolution, batch normalization, and Leaky Rectified Linear Unit (ReLU) function was regularly used. Batch normalization was used on the generator and discriminator but not in the network's last layer. The generator's initial layer was the full connection layer, and the final layer of convolution was activated by the hyperbolic tangent activation function (Tanh).

### 3.3. W-Net Based Segmentation

After the pre-processing, W-net-based segmentation was implemented to obtain the segmentation maps of the cell nucleus. This network preserves the localization and content information using the decoding and encoding paths. Moreover, edge information is preserved to sharpen the image and maintain consistency in segmentation. As an advancement of the U-Net, this network was planned. Thereafter, a single autoencoder was implemented by joining two U-Net topologies together. In each u-net, a contracting (encoder) and an expansive path- (decoder) based structure was implemented.

The first component of the W-net was the contracting path, which comprised a series of blocks. The batch normalization layers and three three-layer convolutional layers interspersed with ReLUs were the blocks' essential components. To create a single convolutional block, this fundamental component was taken into account twice. By using $2 \times 2$ layers of max pooling, the blocks were joined. The critical target information could be preserved, and the number of parameters could be decreased using max pooling. The convolutional layer's kernel count was 8 in the expanding path, rising from 8 to 128 in the contracting path.

The decoder portion was the second-wide path. Layers of convolution and up sampling made up its structure. In the encoder part, the input was downscaled once, and in the decoder part, it was upscaled four times. Thus, the final feature maps of the upsampling process were the same size as the input images. To recover the lost data during the max pooling and convolution processes, the feature maps from the encoder path were concatenated with the equivalent feature maps from the decoder component. This second portion

is equivalent to the first, but in the first U-Net section, the outcome of the units located at the same level and the outcome of the top pooling layers are combined.

Following the upsampling of the decoder's last unit and the final combination of the encoder's initial unit, there was an additional block identical to all the others. In that block, the last layer, a softmax activation function, and a 1×1 convolutional layer was employed to match the desired number of classes and feature maps. This method combined total-variation loss (CT-loss) and cross-entropy loss (CEL).

$$Loss = L_{cr-etrp} + L_{total-var} \tag{4}$$

$$L_{cr-etrp} = L\big(sr'_n, pc_n\big) = -\sum_i^K pc_i \log(sr'_i) \tag{5}$$

$$L_{total-var} = L(\{sr\})'_n = \sum_\xi^{W-1} \sum_\eta^{H-1} \left\| sr'_{\xi+1,\eta} - sr'_{\xi,\eta} \right\| \left\| \left\| sr'_{\xi+1,\eta} - sr'_{\xi,\eta} \right\| 1 \tag{6}$$

Here, W and H represent the width and height of an input image, respectively. The pixel value at that location in the standardized segmentation map $\{pc_n\}$ denotes the pseudo segmentation mask formed by the index that maximizes the value of the measured segmentation map; $\{sr'_n\}$ indicates the sample n's normalized segmentation map. This CT loss helps reduce time and utilize memory. Due to the characteristics of the total-variation loss, the segmentation mask can also be compressed significantly, negating the need for post-processing.

### 3.4. GhostNet-Based Feature Extraction

After the nucleus segmentation, a GhostNet-based deep learning technique was implemented to extract the features from the image. GhostNet suggested a creative Ghost module that produced more feature maps via affordable operations. This fundamental neural network unit could create many image features with fewer inputs and computations. There are two aspects to this module's implementation. To create feature maps with more channels, GhostNet first performed the standard convolutional calculation. Next, it performed a simple operation to create more feature maps. Finally, it concatenated several feature maps to create a new output.

Ghost bottleneck was the fundamental part of GhostNet, which included two ghost modules. The process of creating M feature maps in the ghost modules can be represented as

$$Y = X * f + b \tag{7}$$

Here, the width and height of the input are denoted by w and h, respectively; the number of channels is indicated by c; the bias term is denoted by b; and the convolution operation is represented by *; $f \in R^{c \times k \times k \times m}$, $X \in R^{h \times c \times w}$ is the convolution kernel of this layer. The size of the convolution kernel f is k*k.

Initially, the W × H × C size-based input feature map is downscaled with regular convolution. A cheap linear process is then used on the W′ × H′ × C feature map with a k × k tiny kernel convolution operation, which immediately produces a significant amount of additional ghost features. At last, the outcomes of these two processes are combined to produce an outcome feature map of size W′ × H′ × c that is comparable to the original. The linear transformation and regular convolution used in Ghost Module allow for a more excellent preservation of the original features.

The stride rate of each stage's final bottleneck is set to 2. The stride rate of other ghost bottlenecks is set to 1. Lastly, the feature map is transformed into the final 1280-dimensional feature vector using the global average pooling and convolutional layer. The Ghost module's processing cost is considerably less than that from traditional convolution directly.

*3.5. ResNeXt-Based Classification*

An improved version of ResNet was called ResNeXt. In this network, a parallel stacking block with the same topology was used in place of a three-layer convolutional block. One convolutional layer, four ResNeXt block structures, two pooling layers, one fully-connected layer, and one Softmax classifier were all included in the ResNeXt network. Deep residual networks had cardinality and were made up ResNeXt blocks. They modified the leftover block using the split-transform-merge technique, which led to branching routes inside a cell. With the skip connection path, the ResNeXt block's output was provided. The residual networks' depth grew in an orthogonal manner, and the ResNeXt block's cardinality was determined by the quantity of branching routes it contained. The ResNeXt block can be described mathematically as follows:

$$OP = a + \sum_{i=1}^{ca} \tau_i(a) \tag{8}$$

Here, the arbitrary conversion is denoted by $\tau_i$, cardinality is denoted by *ca*, *OP* represents the output, and the input from the preceding layer is denoted by *a*.

Each ResNeXt block consists of a shortcut connection and three convolutional layers. The three different types of convolutional layers are group convolution, convolution in sequence, and convolution. With the exception of the last convolutional layer, which was followed by a BN (batch normalization) layer, the ReLU activation function was employed to boost the network's nonlinearity after each convolutional layer. The final convolution layer's output was mixed with the input characteristics provided by the shortcut, and it was then activated using the ReLU activation algorithm. The group convolution stride of the ResNeXtBlock1 module was 2. When compared to the size of the input feature map, it could cut the size of the feature map in half.

Max pooling was used in the pooling layer to expedite training and achieve spatial invariance while preserving accuracy. The fundamental concept behind max-pooling was selecting the most discriminative feature and using it to represent a bunch of features. The neighborhood's highest value was determined during the pooling process by

$$v_{i,L}^{x,y} = \frac{max}{m \in [0, m_i - 1], n \in \left[0, n_i - 1\right]_{(i-1),L}^{(x+m),(y+n)}} \tag{9}$$

Here, $m_i, n_i$ is the kernel size, and L indexes the feature map in the $(i-1)$th convolution layer.

The residual learning module's feature map was transformed into a 128-dimensional feature vector and was sent to the fully connected layer using global average pooling. The softmax classifier was then used to perform the final classification.

Parameter Optimization Using the WHO Technique

The hyperparameters of the proposed framework have been optimized in this study using the WHO Algorithm. The fundamental justification for choosing this method is that, as compared to other kinds of metaheuristic algorithms, it is too new. Additionally, its outcomes for the benchmark functions based on the study also yield improved outcomes. This prompts us to employ this metaheuristic method to raise the proposed technique's effectiveness. The Wildebeests' habit of food-seeking serves as the model for the WHO algorithm. Wildebeests are active, sociable mammals that search for food sources. Males compete in sex challenges with rivals to attract females for mating.

Populations (wildebeests) were randomly initialized as candidates at the beginning of the WHO algorithm. In between lower ($X_{min}$) and upper ($X_{max}$) borders, the population was constrained, i.e.,

$$X_i \in [X_{min}, X_{max}] \tag{10}$$

Here, I = 1, 2 . . . N.

The wildebeest subsequently employed the milling method of locomotion. Continuing to look for the optimal position while considering a fixed number (n) as the little random mobility dependent on the location was how this phase was represented. The competitors in position X had employed a random phase $Z_n$ that ought to routinely search for the little random phase spots. The length was adjustable and was determined by the size of the contestants' random steps. Thus, the localized experimental phase $Z_n$ was produced using the following formula:

$$Z_n = X_i + \varepsilon \times \theta \times v \tag{11}$$

Here, a random unit vector is represented by $v$; a random uniform value between 1 and 0 is denoted by $\theta'$; the $i$th candidate number is denoted by $X_i$; and the learning rate is denoted by $\varepsilon$.

After evaluating a fixed number (n) of minor random possibilities, the wildebeest adjusted its location to obtain an ideal random position. It is described in the following Equation (12).

$$X_i = \alpha_1 \times Z_n^* + \beta_1 \times (X_i - Z_n^*) \tag{12}$$

Here, the local movement of the candidates is instructed by the $\alpha_1$ and $\beta_1$ leader variables.

Modeling the wildebeests' swarm behavior was the final step. This was mimicked after placing other contestants in a spot with an appropriate food source, such as

$$X_i = \alpha_2 \times X_i + \beta_2 \times X_h \tag{13}$$

Here, $\alpha_2$ and $\beta_2$ stand in for the leader variables to direct the crew's local movement, and $X_h$ denotes a random candidate.

$$X_i = X_i + \theta \times (X_{max} - X_{min}) \times \overline{v} \tag{14}$$

Here, the random unit vector is denoted by $\overline{v}$.

Simulating busy areas was another phrase used in the algorithm. When the grassland had vast productivity, there was a population. This concept is called individual pressure. This term is used to complete a task, and the best contender uses the following Formula (15) to destroy other contenders.

$$if \left( \|X^* - X_i\| \right) < \eta, \left( \|X^* - X_i\| \right) > 1 \tag{15}$$

$$\text{Then } X_i = X^* + \varepsilon \times \hat{n} \tag{16}$$

where $\eta$ denotes a threshold to prevent congestion in the location, and $\hat{n}$ indicates the number of accessible sections near the ideal solution point.

The swarm's social memory, which was simulated in the last stage to provide better placements, was determined by the following equation:

$$X = X^* + 0.1 \times \hat{v} \tag{17}$$

Finally, the best optimal value of this algorithm was initialized to the hyperparameters of the classifier. The optimized values are a learning rate of 0.001, a weight decay rate of $1 \times 10^{-6}$, a momentum of 0.8, a batch size of 8, and a dropout rate of 40% with L2 regularization.

### 3.6. Computational Complexity

In Ghostnet, the overall time required to generate a network is O (w). Here, W is the number of weights. In w-net, the computational complexity is O (n), where n is the number of layers. In ResNeXt, the computational complexity is $O\left(\sum_{j=1}^{k} x_{j-1} \cdot p_j^2 \cdot x_j y_j^2\right)$. Here, the output feature map's spatial size is denoted by yj; the filter's spatial size is denoted by pj; and the number of kernels and convolutional layers are denoted by x and k, respectively. The complexity of the WHO algorithm is O (p $\times$ m). Here, m and n denote the population

and problem dimension, respectively. Finally, the overall computational complexity of the network is $O\left(\left(\sum_{j=1}^{k} x_{j-1}.p_j^2.x_j y_j^2\right) n \times w \times m\right)$. The total time taken for the proposed approach is 1.95 s.

## 4. Result and Discussion

The results from the proposed model are presented in this section. The experimental investigation for this paper was conducted on a personal computer with an Intel Core (i7) 8700U processor (Intel, Santa Clara, CA, USA) operating at 3.20 GHz, NVIDIA GeForce-GTX 1050 Ti graphics (NVIDIA, Santa Clara, CA, USA) totaling 4 GB, and 16 GB of main memory. Python 3.7 and its associated libraries were used to create the software. To increase the learning capacity of the network and avoid overfitting and network degradation, 30% of the data was reserved for testing and 70% for training. The hyperparameters optimized by the WHO algorithm are a learning rate of 0.001, a weight decay rate of $1 \times 10^{-6}$, a momentum of 0.8, a batch size of 8, and a dropout rate of 40% with L2 regularization. The model's training was set to 100 epochs to ensure learning stability. The training and learning curves converged after 50 epochs, suggesting that the model has stabilized.

### 4.1. Dataset Description

In our experiments, three different datasets were employed for performance evaluation. They were BCCD, LISC, and the single-cell morphological dataset (immature WBCs). The brief description of these datasets is explained as follows, and the properties are given in Table 2.

**Table 2.** Properties of three datasets.

| Dataset | Pixel Size | Number of WBC | | Total Number of Images | Staining | Microscopic and Zoom | Camera |
|---|---|---|---|---|---|---|---|
| BCCD | 320 × 240 | Neutrophil | 3123 | 12,444 | Gismo-right | Regular light microscope Zoom: 100× | CCD color camera |
| | | Monocyte | 3098 | | | | |
| | | Lymphocyte | 3103 | | | | |
| | | Eosinophil | 3120 | | | | |
| LISC | 720 × 576 | Neutrophil | 50 | 242 | Gismo-right | Axioskope40 Zoom: 100× | Sony-SSCDC50AP |
| | | Monocyte | 48 | | | | |
| | | Lymphocyte | 52 | | | | |
| | | Eosinophil | 39 | | | | |
| | | Basophils | 53 | | | | |
| Single Cell morphological dataset | 400 × 400 | Myelocyte | 42 | 3517 | Papanicolaou stain | M8 digital microscope/scanner | - |
| | | Metamyelocytes | 15 | | | | |
| | | Bilobed Promyelocytes | 18 | | | | |
| | | Myeloblast | 3268 | | | | |
| | | Promyelocyte | 70 | | | | |
| | | Monoblast | 26 | | | | |
| | | Erythroblast | 78 | | | | |

**BCCD dataset:**

The Blood Cell Count Detection (BCCD) dataset [27] is publicly available on Kaggle, an online community platform for data scientists and machine learning enthusiasts. It contains cell type designations in a CSV file and 12,453 JPEG images of leukocytes. Neutrophil, monocyte, lymphocyte, and eosinophil are the four cell types included in this collection. In contrast to the 207, 21, 33, and 88 original pictures, there are 3123, 3107, 3103,

and 3120 augmented pictures, respectively. The dataset excludes basophils because they typically comprise less than 1% of all leukocytes.

**LISC dataset:**

Tehran University's Hematology-Oncology Research Centre (Iran) and BMT Research Center of Imam Khomeini hospital in Tehran, Iran provided the LISC dataset [28]. It contained 250 photos of ground truth as well as hematological images obtained from 400 participants on 100 microscopic images. The acquired photos are 720,576 pixels in size and the .bmp file type. Medical experts categorize the dataset into five subcategories using multiple photos, including fifty-three basophils, thirty-nine eosinophils, fifty neutrophils, forty-eight monocytes, and fifty-two lymphocytes.

**Single-cell morphological dataset:**

The single-cell morphological dataset (AML Cytomorphology LMU) [29] used in this study included WBCs from patients with AML and from healthy people. The dataset was created by the Munich University Hospital, using 18,365 single-cell images identified by experts and gathered from peripheral blood samples of 100 AML patients and 100 healthy individuals between 2014 and 2017. There were 15 different single-cell image categories created from the collection. In that, seven classes were leukemic WBC myelocytes, metamyelocytes, bilobed promyelocytes, myeloblasts, promyelocytes, monoblasts, and erythroblasts, whereas the rest were normal WBCs. Based on a recognized morphological classification, professional analyzers assessed the cancer and non-cancerous WBCs.

*4.2. Segmentation Results*

We employed three standard metrics—Dice Similarity Coefficients (DSC), Mean Intersection Over Union (mIoU), and Misclassification Error (ME)—to assess the effectiveness of the proposed segmentation technique statistically. These metrics are described in the following equations:

$$DSC = \frac{2\left|GR_f \cap PR_f\right|}{\left|GR_f\right| + \left|PR_f\right|} \tag{18}$$

$$mIoU = \frac{1}{2}\left(\frac{|GR_b \cap PR_b|}{|GR_b \cup PR_b|} + \frac{\left|GR_f \cap PR_f\right|}{\left|GR_f \cup PR_f\right|}\right) \tag{19}$$

$$ME = 1 - \frac{|GR_b \cap PR_b| + \left|GR_f \cap PR_f\right|}{\left|GR_f\right| + |GR_b|} \tag{20}$$

Here, $GR_b$ and $GR_f$ represent the non-WBC and WBC areas in the ground truth, respectively. In contrast, $PR_b$ and $PR_f$ represent the non-WBC and WBC areas in the expected segmentation findings. However, the lowest values of ME and the greatest values of Dice and mIoU imply superior segmentation results.

4.2.1. Multi-Class Segmentation on BCCD Dataset

The segmentation outcomes for the suggested technique are shown in Figure 2. To locate the WBCs, the nucleus must also be segmented. However, the nucleus characteristics, colors, and shapes differ among different databases. Therefore, the segmentation of the WBC pictures was carried out using an effective W-net-based deep learning approach. The segmented images produced by the suggested method from the BCCD dataset were flawless. Even when the pattern borders were uneven, the segmentation correctly retrieved the nucleus areas.

Figure 3 displays the quantitative outcomes of the suggested approach using the BCCD dataset. From Figure 3, it is observed that the proposed approach outperformed every performance measure for all classes. Therefore, it is concluded that our segmentation algorithm successfully segments the photos/images on the BCCD dataset. Particularly, eosinophils and monocytes achieved better results than the other classes, with 99.56% and 99.12% DSC, respectively. On the other hand, neutrophils and lymphocytes achieved the highest ME compared to the other classes. As the eosinophils and segmented neutrophils had comparable attribute values, except for the nuclei' ruggedness, a small misclassification occurred. However, the overall segmentation result of the proposed approach for all classes was satisfactory.

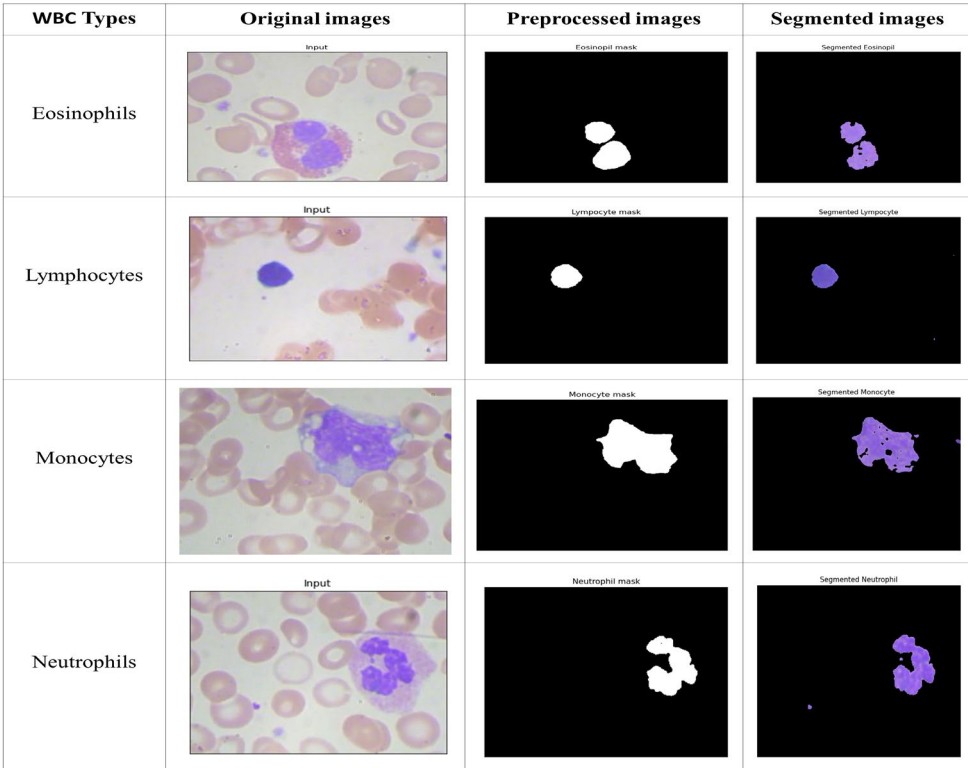

**Figure 2.** Visualization result on BCCD dataset.

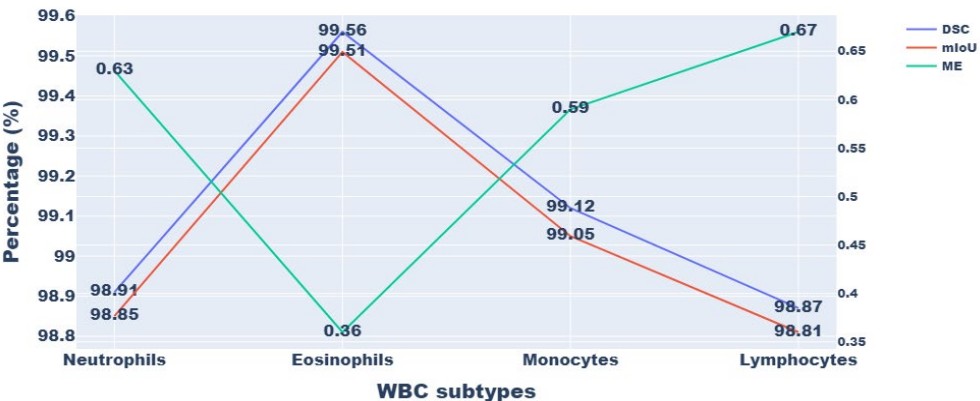

**Figure 3.** Segmentation results on the BCCD dataset.

4.2.2. Multi-Class Segmentation on LISC Dataset

Segmentation of the five main WBC types—neutrophil, monocyte, lymphocyte, eosinophil, and basophil—is illustrated in Figure 4. It helped to understand how the proposed method could give more accurate results without introducing extra artifacts and

how these photos were closer to the ground truth pictures. The approach proposed in this study has the benefit of utilizing two U-net structures, which optimizes the boundary of the segmented cells by fusing low-level and high-level data to improve WBC localization and yield improved results.

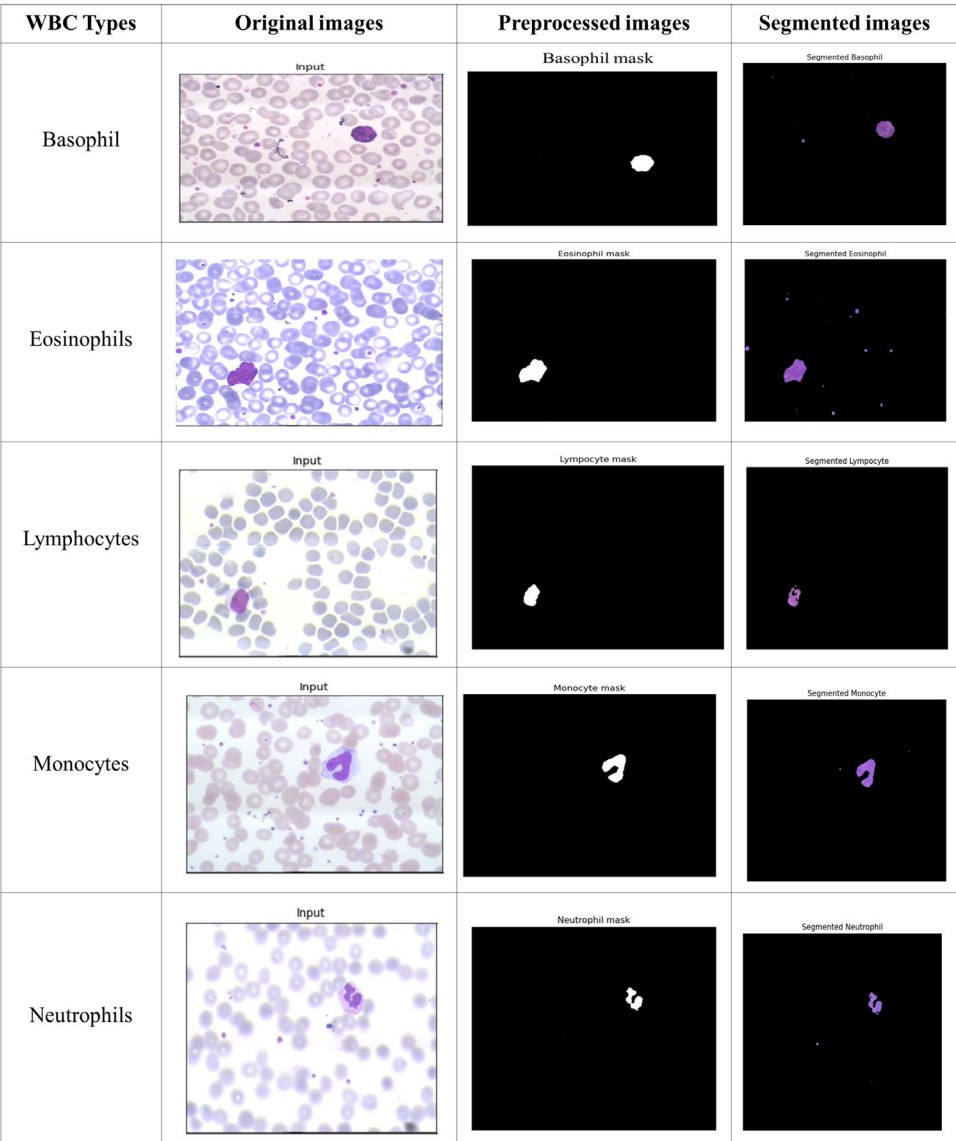

**Figure 4.** Visualization result of five types of WBC on the LISC dataset.

Figure 5 displays the segmentation outcome (quantitative) for the five categories of WBC for the LISC dataset. Here, we can observe that the average DSC for segmentation is >99% for all cell types, with the exception of lymphocytes and monocytes. The monocytes and lymphocytes in the LISC achieve DSCs of 98.90% and 98.88%, respectively. In addition, the precision and DSC of the recommended segmentation method have minimum standard deviations, demonstrating that it consistently performs well for different cells in the dataset. The suggested method also offers low false- and leak-detection ratios and performs well with boundaries that have been traced.

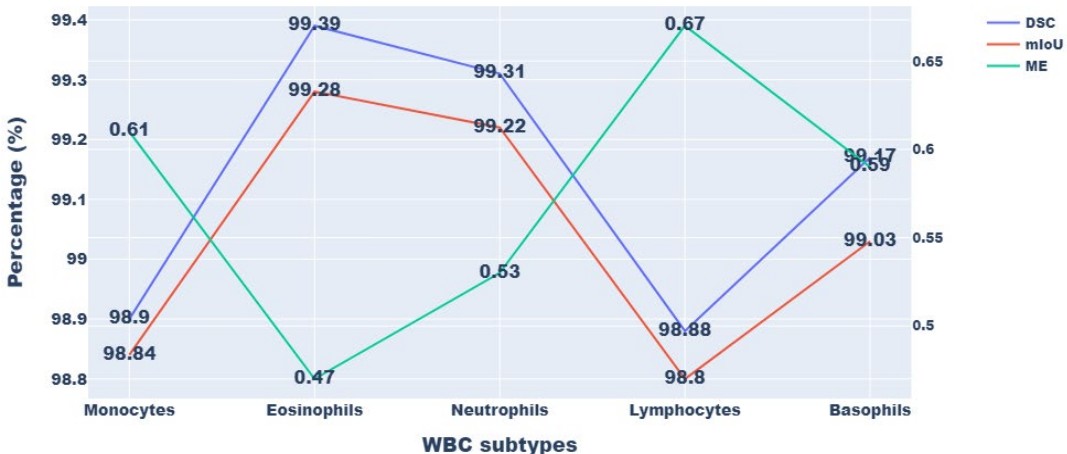

**Figure 5.** Segmentation results on the LISC dataset.

### 4.2.3. Multi-Class Segmentation on Single Cell Morphological Dataset

Figure 6 shows the visualization result of immature WBC segmentation on a single-cell morphological dataset.

| WBC Types | Original images | Preprocessed images | Segmented images |
|---|---|---|---|
| Erythroblast | | | |
| Metamyelocyte | | | |
| Monoblast | | | |
| Myelocyte | | | |
| Myeloblast | | | |
| Promyelocyte (bilobled) | | | |
| Promyelocyte | | | |

**Figure 6.** Visualization result on single-cell morphology dataset.

Figure 7 shows the quantitative outcomes. Among the atypical WBCs, myeloblasts, the most important cell type for the diagnosis of AML, were particularly well-classified by the model. The model's classification performance for myeloblasts was 99.45% DSC and

99.36% mIoU. In addition, the method was successful in classifying additional blast cells, such as erythroblasts and monoblasts.

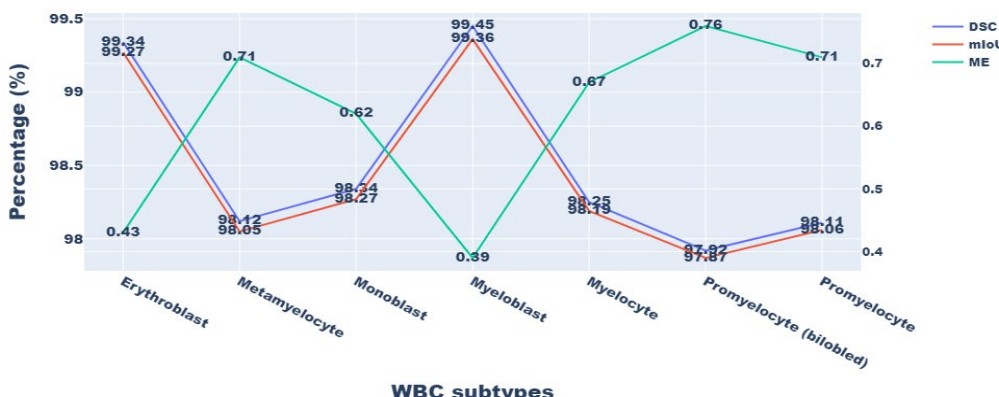

**Figure 7.** Segmentation results of various WBC subtypes on the single-cell morphology dataset.

Segmenting myeloblasts is easier than segmenting promyelocytes (bilobed) because the dataset of promyeloblasts comprises only a tiny number of instances for promyelocytes (bilobed). The samples of this class are enlarged due to the suggested data augmentation technique, known as DCGAN, and improved results are obtained, such as 97.92% DSC, 97.87% mIoU, and 0.76% ME.

In the initial stage of the metamyelocyte stage, these WBC types are misclassified as myelocytes (Figure 7). Similarly, in the initial phase of promyelocytes, they were misclassified as myeloblasts. It could be because metamyelocytes and promyelocytes (bilobed), unlike erythroblasts and monoblasts, are in the intermediate phases of the myelocyte, a very complex process that is prone to categorization error.

*4.3. Classification Results*

Accuracy, precision, recall, and F1-score are the evaluation measures used in this study for the categorization of WBC subtypes. The following formulae are used to determine these metrics:

$$Accuracy = \frac{TrP + TrN}{TrP + TrN + FsP + FsN} \tag{21}$$

$$Precision = \frac{TrP}{TrP + FsP} \tag{22}$$

$$Recall = \frac{TrP}{TrP + FsN} \tag{23}$$

$$F1 - Score = \frac{2 \times precision \times recall}{precision + recall} \tag{24}$$

Here, False Negative (FsN) indicates the number of blood cell types that were improperly identified. FsP counts the number of cells that were mistakenly classified as the wrong type of blood cell, while True Positive (TrP) indicates the number of blood cell types that were correctly separated. The number of cells correctly detected as not being the target blood cell type is known as True Negative (TrN).

4.3.1. Multi-Class Classification of BCCD Dataset

The classification accuracy, precision, recall rate, and F1 score for various cell types in the BCCD dataset are shown in Table 3. The categorization performance measures in the table show the selected classifier's capacity to distinguish between various categories on the BCCD dataset. The proportion of successfully identified images is indicated by the sensitivity, often known as the "recall". The proposed method, except for eosinophils, delivers above 99% recall on the BCCD dataset. Eosinophils have a 98.95% recall rate.

However, these subtypes achieved better outcomes in terms of precision (99.16%) and accuracy (99.04%).

**Table 3.** Quantitative results of multi-class classification on BCCD dataset.

| WBC Subtypes | Accuracy | Precision | Recall | F1-Score |
|---|---|---|---|---|
| Monocytes | 99.12 | 99.23 | 99.07 | 99.15 |
| Eosinophils | 99.04 | 99.16 | 98.95 | 99.02 |
| Neutrophils | 99.57 | 99.64 | 99.51 | 99.57 |
| Lymphocytes | 99.85 | 99.89 | 99.81 | 99.87 |

The F1 score is the balanced mean of the classifier's recall and precision, in which the precision considers erroneous positives. Precision and recall metrics typically have to be traded off. The highest F-measure indicates that recall and precision have similarly high values. Table 3 shows that the lymphocytes class achieves the most excellent F1-score value (99.87%) and superior precision (99.89%) and recall (99.81%) values.

### 4.3.2. Multi-Class Classification on LISC Dataset

Table 4 displays the suggested model's performance on the LISC dataset. The performance of the proposed classifier gave slightly less accurate results in the lymphocyte and monocyte subtypes, as shown in Table 3, because these classes discriminate poorly and have comparable cell nuclei. In comparison to these categories, the suggested technique successfully identified neutrophils, eosinophils, and basophils with significant F1 scores of 99.34%, 99.47%, and 99.21%, respectively.

**Table 4.** Multi-class classification on the LISC dataset.

| WBC Subtypes | Accuracy | Precision | Recall | F1-Score |
|---|---|---|---|---|
| Monocytes | 98.92 | 99.11 | 99 | 99 |
| Eosinophils | 99.47 | 99.58 | 99.42 | 99.50 |
| Neutrophils | 99.34 | 99.41 | 99.37 | 99.39 |
| Lymphocytes | 98.87 | 98.98 | 98.91 | 98.94 |
| Basophils | 99.21 | 99.34 | 99.27 | 99.30 |

Additionally, the model had 98.87% accuracy, 98.98% precision, 98.91% recall, and a 98.94% F1-score when it came to differentiating atypical lymphocytes from other WBCs. Given the limited number of atypical cells in the dataset, our results and accuracy data are very good. A cell can be reinfected with bacteria, viruses, or parasites. Typical lymphocytes have more stable nuclei than monocytes, despite having similar cytoplasm size and volume. Therefore, the exact categorization of atypical cells is a challenging issue.

### 4.3.3. Multi-Class Classification on Single Cell Morphology Dataset

Table 5 summarizes the findings of the multi-class categorization of immature WBCs. The proposed model was tested on all classes and had precision and recall values above 98%, except for the promyelocyte (bilobed) class. With average precision used as the scoring criterion during model construction, the improved model achieved precision above 97% for all classes (see Table 5). For the myeloblast class, which is the most prevalent immature leukocyte in patients with AML, the model achieved 99.56% precision and 99.32% recall.

The sensitivity and precision of the suggested model summarized by the F-score are shown in Table 5. Myeloblasts and erythroblasts received the best scores, while promonocytes and metamyelocytes achieved the worst score. This was due to the difference in properties between myeloblasts and erythroblasts, which originate from two distinct myelopoiesis branches. Additionally, the availability of myeloblast images helped develop more specific features and produce more exact outcomes. In contrast, promono-

cytes and metamyelocytes were prone to misclassification since they are in successive stages of myelopoiesis.

**Table 5.** Multi-class classification on single-cell morphology dataset.

| WBC Subtypes | Accuracy | Precision | Recall | F1-Score |
|---|---|---|---|---|
| Erythroblast | 99.62 | 99.73 | 98.67 | 99.2 |
| Metamyelocyte | 98.15 | 98.23 | 98.2 | 98.21 |
| Monoblast | 98.45 | 98.49 | 98.47 | 98.48 |
| Myeloblast | 99.51 | 99.56 | 99.32 | 99.44 |
| Myelocyte | 98.35 | 98.38 | 98.47 | 98.42 |
| bilobled | 97.99 | 97.98 | 98 | 98 |
| Promyelocyte | 98.21 | 98.26 | 98.33 | 98.29 |

### 4.3.4. Comparison of the Proposed Methodology with Existing Techniques

Table 6 compares and evaluates the classification results for the proposed methodology with other advanced methods. The ResNeXt with a WHO algorithm was utilized for classification purposes. The optimization strategy utilized in the suggested approach will increase classification accuracy while also enhancing algorithm performance.

**Table 6.** Comparison of the proposed approach with existing techniques on three datasets.

| Dataset | Techniques | Accuracy | Precision | Recall | F1-Score |
|---|---|---|---|---|---|
| BCCD | Resnet-densenet-SCAM [30] | 88.44 | 90.84 | 88.45 | 88.73 |
| | WBC-AMNet [31] | 89.22 | 90.72 | 89.22 | 89.47 |
| | Cubic SVM [32] | 98.44 | - | - | - |
| | DRFA-Net [33] | 95.87 | 96.13 | 92.94 | 94.51 |
| | Proposed | 99.24 | 99.48 | 99.33 | 99.4 |
| LISC | K-nearest neighbor [34] | 97.8 | - | - | - |
| | Alexnet [35] | 97.21 | 72.75 | 89.6 | - |
| | CNN [36] | - | 93.42 | 96.27 | 94.73 |
| | SVM [37] | 92.21 | 92.65 | 92.44 | 92.44 |
| | Proposed | 99.16 | 99.28 | 99.19 | 99.22 |
| Single-cell morphology dataset | deep convolutional autoencoder (DCAE) [22] | 93.12 | 67.35 | 81.65 | - |
| | XGBoost [38] | 97.57 | 98.48 | 97.16 | 97.82 |
| | Random forest [24] | 93.4 | 86.68 | 92.26 | - |
| | Deep convolutional Neural network (DCNN) [39] | 98.27 | 96.95 | 98.04 | - |
| | Proposed | 98.61 | 98.66 | 98.49 | 98.57 |

From Table 6, it is observed that the performance of Resnet-densenet-SCAM on the BCCD dataset was very low (88.44% accuracy) compared to other techniques. They did not use any segmentation technique. For feature extraction, the entire input image was used. As a result, undesirable traits were also extracted from the image. Moreover, the loss function used in this paper ignores the variations among the samples, which leads to poor performance.

In the LISC dataset, the performance SVM was slightly decreased compared to that of other techniques. This was because the images in the LISC dataset contains a vast range of hue and intensity fluctuations. To tackle this problem, the pre-processing techniques were not implemented in [37] because it leads to misclassification.

In the single-cell morphology dataset, four existing techniques were used for the comparison with the proposed approach. In that, the performance of the deep convolutional autoencoder (DCAE) and random forest was lower than the other techniques. Random forest technique contains many significant hyper parameters. The manual tuning of these hyper-parameters reduces the performance of the system. Moreover, the entire raw input image is processed in DCAE [22] for classification. They do not perform any segmentation

process. As a result, undesirable traits are also extracted from the image, which slows down the performance of the classifier.

Compared to all the other techniques, the proposed method produced the best results for the datasets for BCCD, LISC, and single-cell morphology with 99.24%, 99.16%, and 98.57% accuracy, respectively.

The following are reasons for the greater accuracy of the proposed method:

(a) A better feature extraction technique based on GhostNet obtained low and high-level characteristics of the image.

(b) Use of a ResNext-based good classification technique that incorporated the WHO algorithm for parameterization that worked effectively with these characteristics.

(c) A superior segmentation depending on W-net assists in identifying WBCs and yields accurate information relevant to the structure of the nucleus for each category of WBCs.

Finally, from all the above observations, it is concluded that the workflow of algorithms in this study showed the best performance compared to that of other existing techniques.

### 4.3.5. Ablation Study

To demonstrate the impact of our suggested framework, ablation experiments are presented in this subsection. The proposed framework contains five modules: pre-processing, data augmentation based on DCGAN, segmentation based on W-net, feature extraction based on GhostNet, and classification based on ResNext with WHO-based hyperparameter optimization. The above-mentioned testing results demonstrated that the suggested approach can deliver a powerful performance. In order to examine the impact of each component in the suggested framework, we further implemented the ablation experiment in this section.

In order to achieve this, networks with different module combinations were constructed. Table 7 presents the quantitative results. It only achieves 98.93% accuracy, 99.12% precision, 99% recall, and 99.06% F1-score in the BCCD dataset without the W-net phase. The impact of data augmentation (DCGAN) on the outcomes is another factor. It achieves just 99% accuracy without the DCGAN.

**Table 7.** Impact of the various modules on proposed approach.

| DCGAN Data Augmentation | W-Net Segmentation | WHO Hyperparameter Optimization | Accuracy | Precision | Recall | F1-Score |
|---|---|---|---|---|---|---|
| BCCD dataset | | | | | | |
| | ✔ | ✔ | 99 | 99.15 | 99.06 | 99.11 |
| ✔ | | ✔ | 98.93 | 99.12 | 99 | 99.06 |
| ✔ | ✔ | | 99.06 | 99.20 | 99.12 | 99.16 |
| ✔ | ✔ | ✔ | 99.24 | 99.48 | 99.33 | 99.4 |
| LISC dataset | | | | | | |
| | ✔ | ✔ | 98.96 | 99.10 | 99 | 99.05 |
| ✔ | | ✔ | 98.87 | 98.98 | 98.91 | 98.94 |
| ✔ | ✔ | | 99.04 | 99.17 | 99.09 | 99.13 |
| ✔ | ✔ | ✔ | 99.16 | 99.28 | 99.19 | 99.22 |
| Single-cell morphology dataset | | | | | | |
| | ✔ | ✔ | 98.52 | 98.58 | 98.43 | 98.50 |
| ✔ | | ✔ | 98.43 | 98.50 | 98.34 | 98.42 |
| ✔ | ✔ | | 98.57 | 98.62 | 98.45 | 98.53 |
| ✔ | ✔ | ✔ | 98.61 | 98.66 | 98.49 | 98.57 |

## 5. Conclusions

The classification of WBCs is one of the most critical problems in the medical field. An increasing number of infection cases and the challenges associated with the early detection of these infections make it crucial to properly classify WBCs. Within the context of this paper, an effective GhostNet- and an optimized ResNeXt-based feature extraction and classification were conducted with the aid of W-net-based segmentation to classify matured and immature WBCs. With accuracy rates of 99.24%, 99.16%, and 98.61% for the BCCD, LISC, and single-cell morphology datasets, respectively, the proposed model surpassed previous approaches. Furthermore, the segmentation effectiveness of the suggested strategy is equally satisfactory and produced one of the finest outcomes. It shows that the suggested method is effective for identifying mature and immature WBCs in both clinical and diagnostic labs. In the future, we want to employ the suggested architecture to accurately differentiate leukemia cells from other forms, such as acute lymphoblastic leukemia (ALL).

**Author Contributions:** Conceptualization, S.S.R.B.; methodology, S.R.B.; writing—original draft preparation, S.S.R.B.; supervision, S.R.B. All authors have read and agreed to the published version of the manuscript.

**Funding:** This research received no external funding.

**Institutional Review Board Statement:** Not applicable.

**Informed Consent Statement:** Not applicable.

**Data Availability Statement:** Publicly available.

**Conflicts of Interest:** The authors declare no conflict of interest.

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
