# Peer review of "Ghost-ResNeXt: An Effective Deep Learning Based on Mature and Immature WBC Classification"

_applsci, doi:10.3390/app13064054_

Round 1

Reviewer 1 Report

The manuscript presents a deep learning model to categorize the WBC from the images of peripheral blood smears.

1. In section 4, give the reason in selecting 70% and 30% for training and testing respectively.

2. Authors did not present the computational time and complexity analysis of the proposed method.

3. The quality of Figures 1, 3, 5, 7 are very poor. Authors are instructed to present high quality (max. dpi) pictures in the manuscript.

4. The full-forms of abbreviations are repeatedly mentioned in the manuscript. For example, WHO, DCGAN. It is instructed to mention the full-forms when they are presented first time, afterwards abbreviations need to be used.

5. There are certain abbreviations whose full-forms are not mentioned. For example, DSC. Authors should ensure that the full-forms are mentioned for all the abbreviations.

6. In section 4.1, the references are not mentioned for the data sets used in the manuscript.

7. Citation of references in the main text are not uniform. For example, [22]. Authors are instructed to maintain uniformity in citations of references.

8. The format of references is not uniform. For example, Ref. [7], [26]. Authors are instructed to give the complete details of the references that includes, Vol. No. issue. No. page numbers, year, and DOI.

9. There are certain errors in English sentence formation, grammar, and spellings. Authors are instructed to read the manuscript carefully and rectify the grammatical errors.

Author Response

Thank you for giving us the opportunity to submit a revised draft of the manuscript “Ghost- ResNeXt: An Effective deep learning based on mature and immature WBC classification” for publication in the journal of “Applied Sciences”. We appreciate the time and effort that you and the reviewers dedicated to providing feedback on our manuscript and are grateful for the insightful comments on and valuable improvements to our paper. We have incorporated most of the suggestions made by the reviewers. Those changes are highlighted within the manuscript. Number wise answers to their specific comments are as follows

Reviewer 2 Report

This paper created a deep learning model for the categorization of WBC. The authors proposed five methods: CLAHE, DCGAN, W-net, GhostNet and ResNeXt, respectively for image preprocessing, image enhancement, WBC segmentation, feature extraction and classification. In addition, the authors used three different datasets to evaluate the performance of the proposed methods. However, I have the following suggestions for authors to consider.

(1)   In Figure 3, Figure 5 and Figure 6, authors applied W-net for segmentation, but there are many other segmentation methods, why do the authors choose W-net? If the authors compared the W-net with other methods, the model could more convincing.

(2)   The author uses DCGAN for image enhancement. Why do authors choose DCGAN? What is the performance of other image enhancement methods.

(3)   In 3.5.1, the author introduced the Wildebeest Herd Optimization (WHO) technique, and did not explain how to use this technique to optimize the hyperparameters of the framework anywhere else in the article.

(4)   In the second paragraph of 4.3.1, the authors mistakenly wrote Table 1 as Table 2.

(5)   In 4.2, the authors mentioned that four standard metrics were used to assess the effectiveness of the proposed segmentation technique statistically, but only three metrics were introduced in the paper. In addition, the DSC metric in Figure 3, Figure 5 and Figure 6 have not been introduced previously.

(6)   Some illustrations in the article are low-resolution and look blurry.

Author Response

Thank you for giving us the opportunity to submit a revised draft of the manuscript “Ghost- ResNeXt: An Effective deep learning based on mature and immature WBC classification” for publication in the journal of “Applied Sciences”. We appreciate the time and effort that you and the reviewers dedicated to providing feedback on our manuscript and are grateful for the insightful comments on and valuable improvements to our paper. We have incorporated most of the suggestions made by the reviewers. Those changes are highlighted within the manuscript. Number wise answers to their specific comments are as follows.

Reviewer 3 Report

In this research, author develop a deep learning model for classifying WBC, including immature WBCs, from pictures. Initial work on leukocyte segmentation was performed using a W-Net-based network. Then, a deep learning framework based on GhostNet is utilized to retrieve the most pertinent feature maps. The data is then put through a classification process based on ResNeXt and WHO (Wildebeest Herd Optimization). Moreover, data augmentation based on a Deep Convolutional Generative Adversarial Network (DCGAN) is used to deal with the asymmetry in the data.

I have appreciated the related work section in this paper, but I suggest you add a table that compares the different works in this section.
In terms of results, I suggest you use another deep learning model other than ResNet.
Table 4, I find that your result is very high compared to other approaches, especially [27]. If it is possible to discuss this theoretically, and expand the discussion of the results in section 4.3.4,

Author Response

(The authors gave the same response as above.)

Round 2

Reviewer 2 Report

(1)    The authors use DCGAN for data augmentation. However, I wonder if other GANs have been considered, such as DAGAN. In addition, I recommend an ablation experiment to verify the necessity of using DCGAN for data augmentation.
(2)    In 4.1, the authors briefly describe the three datasets. However, I suggest you add a table that compares the key information and attributes of the datasets.
(3)    I hope the authors carefully check the punctuation in the paper, such as the end of the first paragraph in the second section, lacking a closing character.

Author Response

(The authors gave the same response as above.)
